# A Novel Method for Pattern Recognition of GIS Partial Discharge via Multi-Information Ensemble Learning

**DOI:** 10.3390/e24070954

**Published:** 2022-07-09

**Authors:** Qianzhen Jing, Jing Yan, Lei Lu, Yifan Xu, Fan Yang

**Affiliations:** State Key Laboratory of Electrical Insulation and Power Equipment, Xi’an Jiaotong University, Xi’an 710049, China; jingqianzhen@stu.xjtu.edu.cn (Q.J.); dianqilu@stu.xjtu.edu.cn (L.L.); xyf9894326@stu.xjtu.edu.cn (Y.X.); yangfan980925@stu.xjtu.edu.cn (F.Y.)

**Keywords:** multi-information ensemble learning, partial discharge, gas-insulated switchgear, pattern recognition

## Abstract

Partial discharge (PD) is the main feature that effectively reflects the internal insulation defects of gas-insulated switchgear (GIS). It is of great significance to diagnose the types of insulation faults by recognizing PD to ensure the normal operation of GIS. However, the traditional diagnosis method based on single feature information analysis has a low recognition accuracy of PD, and there are great differences in the diagnosis effect of various insulation defects. To make the most of the rich insulation state information contained in PD, we propose a novel multi-information ensemble learning for PD pattern recognition. First, the ultra-high frequency and ultrasonic data of PD under four typical defects of GIS are obtained through experiment. Then the deep residual convolution neural network is used to automatically extract discriminative features. Finally, multi-information ensemble learning is used to classify PD types at the decision level, which can complement the shortcomings of the independent recognition of the two types of feature information and has higher accuracy and reliability. Experiments show that the accuracy of the proposed method can reach 97.500%, which greatly improves the diagnosis accuracy of various insulation defects.

## 1. Introduction

Gas-insulated switchgear (GIS) failures are usually caused by internal insulation defects. When there is an insulation defect inside the GIS, due to electric field distortion, local breakdown occurs, and a partial discharge (PD) signal is generated. PD is a precursor to a breakdown that causes insulation degradation. When PD occurs in the GIS, ultrasonic waves and electromagnetic wave signals are generated and radiated to the outside, and the detection of PD can be realized by measuring the sound waves and electromagnetic wave signals. Long-term PD will gradually lead to insulation deterioration or even breakdown [1,2,3]. Therefore, the pattern recognition of PD is not only conducive to the monitoring and forecasting of GIS operation status and the determination of targeted maintenance plans, but also provides an important basis for correctly evaluating the hazard of PD and formulating reasonable disposal measures [4].

At present, in the field of PD pattern recognition, domestic and foreign scholars have carried out a lot of related research work and achieved remarkable results. Liu et al. [5] proposed a novel GIS PD pattern recognition method by using CNN and long short-term memory (LSTM), and the effectiveness of this method is verified by the experimental PD dataset. Wang et al. [6] proposed a MixNet deep learning model to optimize PD pattern recognition and the model is verified by ultra-high frequency (UHF) PD data collected in the experiment. The results demonstrated its superiority over the other traditional methods. Tuyet-D et al. [7] improved LSTM by adding a self-attention block, which has achieved significant performance compared with the traditional diagnostic methods for the phase-resolved PD pattern recognition of GIS.

However, the above studies all use a single signal feature to recognize the PD pattern and cannot make the most of the PD signal. PD is associated with physical phenomena such as electricity, light, sound, and chemical substances. In principle, any of the above-mentioned signals can be used to reveal PD phenomena. In addition to the traditional pulse current method and UHF method, local monitoring methods based on ultrasonic signals have also been proved to be effective and accumulated a rich knowledge system. Compared with abundant electrical monitoring, the acoustic detection method has the advantage of strong anti-electromagnetic interference ability and is especially suitable for GIS PD monitoring in a strong electromagnetic interference environment [8]. Multi-information fusion technology is an inevitable trend of online monitoring. It can comprehensively and intelligently process multi-source information from the same target and obtain more accuracy and comprehensive judgment processing than a single information source, effectively improving its comprehensive recognition ability [9]. Wu Y et al. [10] proposed a multi-information fusion PD pattern recognition method by using Dempster–Shafer (D-S) evidence theory. Based on the statistical data features and image data features of phase-resolved PD and time-resolved PD, the fusion output based on UHF and ultrasonic data recognition models are established respectively, so as to realize the complementary advantages of the two features. Yao Q et al. [11] used UHF and ultrasonic information of GIS insulation defect to construct a multi-source information evaluation model, which successfully improved the reliability of the GIS insulation state evaluation result. Li L et al. [12] established a back propagation neural network (BPNN) diagnosis model based on phase-resolved PD and time-resolved PD respectively preliminary diagnostic results two diagnosis result. The proposed multi-information fusion method combined the characteristics of the two kinds of analysis methods and achieved excellent diagnostic results. 

Although the above methods introduce multi-information fusion technology in the GIS PD pattern recognition task, the diagnosis of PD samples is still limited to the traditional analysis methods based on statistical parameters, and the powerful ability of deep learning to automatically extract features is not exerted. The diagnosis model based on BPNN cannot achieve a deeper network structure, which limits the ability of the network to extract sample features. At the same time, the diagnosis method based on the traditional D-S evidence theory fusion has some abnormal evidence, which may easily lead to conflicts between the evidence, making the fusion diagnosis results inaccuracy or misclassification [13]. Hence, in this paper, a novel multi-information ensemble learning method is adopted to fuse the complementary information provided by the UHF detection method and the ultrasonic detection method to make decisions to improve the reliability of GIS PD pattern recognition.

On the basis of changing the feature information extracted from the traditional single signal for PD pattern recognition, this study explores the effect of maximizing the use of rich insulation state information contained in PD. The PD signal obtained by the comprehensive UHF detection method and the ultrasonic detection method is used as a characteristic quantity to characterize the type of insulation defects in the GIS. First, deep residual convolutional neural network (CNN) is used to automatically extract discriminative features of the two types of PD signals. Then, the multi-information ensemble learning is used to fuse the different feature information of PD at the decision-level, and the difference and complementarity of the two types of feature information are combined to improve the recognition ability of insulation defect types. The specific contributions of this paper are as follows.

(1) The multi-information ensemble learning method is used to fuse the feature information of the UHF signals and the ultrasonic signals at the decision-making level, realizing the complementary advantages of the two types feature recognition, with better fault characterization ability, and can more comprehensively reflect the insulation defects type. The proposed method can make more full use of the rich insulation state information in the recognition results, so that some misjudged samples based on a single fault signal can be fused to obtain correct recognition results, which effectively improves their comprehensive recognition ability.

(2) The deep residual CNN is used to automatically extract the features of PD samples in the form of images, which avoids the tedious process of manually extracting the statistical features of PD signals in traditional statistical parameter analysis methods. At the same time, the skip structure of the residual network can effectively avoid the model degradation problem caused by directly increasing the number of network layers while ensuring high recognition accuracy, and further enhance the performance of the deep CNN.

(3) Experiments verify the effectiveness and reliability of the proposed method. The experimental results show that the accuracy of pattern recognition based on multi-information ensemble learning can reach 97.50%, which is much higher than the recognition results obtained based on a single information source. Meanwhile, the recognition accuracy of various defects has also been significantly improved, indicating that the pattern recognition after fusion by the decision-making layer is more reliable, and its comprehensive recognition ability is effectively improved.

The structure of this paper is organized as follows: Section 2 introduces the basic theory of the experimental method and the experimental platform; Section 3 introduces the overall framework of the proposed method based on multi-information ensemble learning; Section 4 shows the diagnosis result and performance evaluation of the proposed method and compares it with several different methods. Finally, the conclusion is summarized in Section 5.

## 2. The Proposed Method

### 2.1. The Diagnostic Framework Based on Multi-Information Ensemble Learning

The purpose of multi-information fusion is to process multi-source information intelligently and comprehensively from a certain target and obtain a solution that is more accurate and comprehensive than a single source of information for estimation and judgment. This paper proposes a multi-information ensemble learning diagnostic method that fuses ultrasonic signals and UHF signals. The overall framework of the diagnosis network is shown in Figure 1. Firstly, UHF signals and ultrasonic signals of four types of typical defects were obtained through GIS PD experiments. After image preprocessing, the datasets obtained based on UHF and ultrasonic signals were input into a deep residual CNN for preliminary feature extraction of PD in typical insulation defects. Finally, through the ensemble learning, the discriminative features extracted by the deep residual CNN model were fused at the decision-level to form the PD diagnosis result.

### 2.2. The Feature Extraction Based on Deep Residual CNN

With the continuous increase of the network depth, the accuracy of the traditional CNN has been continuously improved. When the network level increases to a certain number, the phenomenon of gradient disappearance and gradient explosion will occur, resulting in poor network training effect and reduced recognition accuracy. Deep residual CNN takes this as a starting point and introduces a structure called shortcut connection, which enables gradients in backpropagation to propagate across one or more layers of the network without exploding or disappearing in layer-by-layer operations. The application of the residual learning module can effectively avoid the problem of gradient explosion and disappearance when the neural network reaches a certain number of layers, and further optimize the performance of the deep network [14].

The basic structure of the deep residual CNN is shown in the Figure 2. Suppose that the input of the residual network structural unit is *x* and the desired output is H(x), i.e., H(x) is the desired complex latent map. If the residual network has been trained to achieve a saturated accuracy or the error of the lower layer is found to increase, the learning goal will be changed to the learning of the identity mapping, so that the input *x* is similar to the output H(x), so as to ensure that the subsequent network levels will not cause a decline in accuracy. The output result H(x) can be represented by the expression H(x)=F(x)+x. When F(x)=0, H(x)=x, which is the identity map. The skip structure of the residual network can effectively avoid gradient disappearance and gradient explosion while ensuring high recognition accuracy, which is suitable for the pattern recognition task of identifying PD defect type. The network structure of the deep residual CNN used in this paper is shown in Figure 3.

### 2.3. Decision-Level Fusion Recognition Based on Ensemble Learning

The decision fusion method comes from the idea of ensemble learning. Ensemble learning is not a single machine learning algorithm, but builds and combines multiple base learners to complete the task [15]. The ensemble learning method obtains an ensemble diagnostic model with better performance in terms of accuracy, generalization, and robustness by combining multiple learning models. The ensemble learning algorithm mainly includes bagging, boosting, and stacking. Stacking algorithm refers to combining the training results of multiple base learners for other algorithms. Simply put, it combines multiple learners in a stacked manner. First, multiple algorithms of the base layer are trained on the complete training dataset and obtain the corresponding output, and then the output of the base layer is used as the input of the next layer algorithm to train and obtain the final result.

Decision-level fusion fuses the classification information of the classifiers according to the corresponding criteria, so as to make the global optimal decision. Feature fusion uses multiple existing feature sets to generate new fusion features through certain rules. It eliminates redundant information caused by the correlation between different feature sets by obtaining the most different information in the original feature set. The whole algorithm is divided into two layers. The first field uses machine learning and deep learning models as base learners and inputs the original data into the base learners for training. The second layer fuses the features obtained by the first-layer base learner to classify faults and obtains the output of the algorithm.

The overall idea of voting is to synthesize the voter’s choice to make the optimal diagnosis. The voting method is widely used in classification problems. It integrates multiple base classifiers through linear combinations, which can integrate the complementary information between the base classifiers and reduce the classification error from a single classifier. Therefore, the voted results are often more accurate than the prediction results of a single classifier. The voting method generally includes majority voting and weighted voting. Among them, the majority voting method follows the principle of minority obeying the majority, and its base classifier only outputs the predicted category, and more than half of them are the final category. Considering that the classification performance of a single classifier for samples is different, the importance of each classifier needs to be evaluated in the voting process, and a larger weight value is given to the strong classifier. The weighted votes of the classification results of various possible outputs from the classifier are calculated, and the classification result corresponding to the highest weighted votes as the decision fusion result is used. This method is called decision fusion based on the weighted voting method. The output of decision fusion based on the weighted voting method is as follows:(1)$$D\left(X\right)=G_{\mathrm{argmax}\sum_{t=1}^{T}v_{t}d_{t}^{j}\left(X\right)}$$
where $v_{t}$ is the weight assigned to the classifier *t*.

For optimal diagnostic performance, the stacking weight set is estimated by minimizing the mean square linear regression. Therefore, the objective function under the two constraints is as follows:(2)$$\Omega =\mathrm{argmin}{\sum_{i=1}^{N}\left[y_{O,i}-\sum_{m=1}^{M}\omega_{m}f_{m,i}\right]}^{2}$$
(3)$$\omega_{m}\geq 0\;m=1,\;2,\;\ldots \;,\;M$$
(4)$$\sum_{m=1}^{M}\omega_{m}=1\;m=1,\;2,\;\ldots \;,\;M$$
where $\Omega =\left\{\omega_{1},\omega_{2},\ldots ,\omega_{M}\right\}$ is the set of weights assigned to the base learners.

Assuming that $p_{t}$ is the probability that the recognition result of the *t*-th classifier is correct, and the outputs of each classifier are relatively independent, there are:(5)$$H\left(X\right)=\log P\left(G_{j}\right)+\sum_{t=1}^{T}d_{t}^{j}\left(X\right)\log\left[p_{t}/\left(1-p_{t}\right)\right]$$
where the weight *v* is proportional to $\log\left[p_{t}/\left(1-p_{t}\right)\right]$, so the setting of the weight coefficient should be proportional to the performance of the classifier.

## 3. Experimental Method

### 3.1. Method Theoretical Analysis

At present, the UHF method is generally used in the field for GIS PD monitoring. The UHF detection technology of PD uses broadband high-frequency sensor to detect the electromagnetic wave signal in the range of 300 MHz~1.5 GHz excited by PD inside the GIS, so as to characterize the physical state of PD inside the GIS. The field application shows that the UHF method has the advantages of suppressing the interference of low-frequency corona signals, wide detection range, and easy field implementation. However, due to the complex electromagnetic interference on site and the low sensitivity of the UHF method for detecting some insulation defects, false alarms and omissions of PD occur. The UHF detection method has certain disadvantages for the detection of free metal PD. The random distribution of amplitude and phase makes it easy to be overwhelmed by noise during the field detection process and cannot be effectively identified [16].

The ultrasonic detection of PD has a very high sensitivity to free metal particles, which is determined by the propagation path of the ultrasonic signal inside the GIS. Usually, the ultrasonic signal generated by PD propagates in the SF_6_ gas first, and then propagates in the metal shell after encountering the coupling interface between the gas and the shell. The ultrasonic signal produced by free metal particles is different from that of PD. The free metal particles jump irregularly inside the cylinder under the influence of the electric field force and gravity, and the ultrasonic signal generated after collision with the shell propagates directly in the metal shell without being attenuated by SF_6_ gas. Therefore, ultrasonic PD detection has extremely high detection sensitivity for free metal particle defects. The free metal particles beat in the GIS due to the action of the electric field force and gravity. The particles directly hit the metal shell, and then pass through the GIS shell to the sensor. The propagation medium of the ultrasonic signal is only the GIS metal shell. The ultrasonic signals generated by other typical insulation defects first propagate through the SF_6_ gas and then pass through the metal shell to the sensor. The propagation of ultrasonic signal in metal relative to its attenuation in SF_6_ gas can be approximated as lossless propagation [17].

### 3.2. Experimental Platform and Data Acquisition

The laboratory has built a 252-kV GIS PD test platform. The experimental schematic is shown in Figure 4 and the test platform is shown in Figure 5. The platform is mainly composed of test transformer, power supply system, GIS cavity, external UHF sensor, and external ultrasonic sensor. The diameter of the high-pressure conduit inside the GIS is 190 mm, and the diameter of the shell is 560 mm, which can be filled with SF_6_ gas of 0~0.8 MPa. The voltage regulator is connected to the low voltage side of the voltage transformer and can induce an applied voltage of 0~110 kV on the high voltage side. The PD signal is recorded by Tektronix DPO7104 high-speed digital storage oscilloscope. The detection frequency band of the ultra-high frequency sensor is 300~1500 MHz, and the detection center frequency of the ultrasonic sensor is 40 kHz.

The common insulation defects in GIS operation were simulated, and four types of typical insulation defect models were designed. They were free metal particle defects (M), metal tip defects (N), floating electrode defects (P), and surface discharge defects (O). For the metal tip defects model, a copper needle with a length of 20 mm was installed on the busbar to simulate the metal tip defect. For the free metal particle defects model, copper particles with a freely movable end radius of 1mm were placed at the bottom of the test cavity. For the floating electrode defects model, a metal sheet with a thickness of 1mm was placed in the middle of two epoxy resin boards with a thickness of 8mm, and the whole piece of material was attached to the high-voltage conductor to make it in a state of potential suspension. The surface discharge defects model was composed of 6cm long copper wire adhered to the insulator surface. Different defect models were individually placed inside the GIS cavity and filled with 0.5 MPa SF_6_ gas. The PD test was carried out on four typical defects by the step-by-step boosting method, and the step-up voltage range of the experiment was 35~110 kV. In order to ensure the diversity of experimental data, multiple experiments were performed by changing the defect position [18,19,20]. The original PD signal collected by the ultrasonic sensor and the UHF sensor is shown in Figure 6 and Figure 7. It is inevitable to face various interferences during the entire PD acquisition process, such as communication interference signals in the laboratory, which have a fixed frequency and frequency bandwidth. In addition, other electrical equipment in the laboratory also generates random pulse disturbances during operation. Therefore, the experimental data already contain a lot of background noise [21,22].

## 4. Results and Analysis

### 4.1. Experimental Setup

Through experiments, 2000 sets of UHF PD data and 2000 sets of ultrasonic PD data were obtained for insulation defect diagnosis. It contained 500 sets of data for each of four types of defects: free metal particle defects, metal tip defects, floating electrode defects, and surface discharge defects. Then, we divided the dataset into test and validation sets. The test set contained 300 sets of PD data for each type of defect, which was used for training the deep residual network model. The remaining PD data was used as a test set to verify the diagnostic performance of the trained model, including 200 sets of data for each of the four types of defects. At the same time, the experiment used TensorFlow 1.8.0 as the framework, while the model training was carried out on a computer equipped with an NVIDIA GeForce RTX 3060 GPU (US), an Intel Core i7-10700F CPU (US), and 12 GB RAM.

### 4.2. Analysis of Diagnosis Results

To verify the performance of the PD pattern recognition method based on multi-information fusion proposed in this paper in the diagnosis of insulation defects on the sample set, the ultrasonic dataset and the UHF dataset were input into the deep residual CNN respectively for diagnosis. To avoid the interference of accidental factors, the average value was taken as the final diagnosis result after ten times of network training. The training accuracy and cross_validation accuracy curve is shown in Figure 8 and the training loss and cross_validation loss curve is shown in Figure 9. Table 1 shows the diagnosis results based on single class dataset and multi-information ensemble learning. It can be seen from Table 1 that under the diagnosis of UHF dataset and ultrasonic dataset, the diagnostic accuracy reached 91.625% and 88.375% respectively. UHF detection method had the best recognition effect for metal tip defects (N) samples. From the UHF image, it can be seen that there was a clear difference between metal tip defects and the other three defects. The recognition accuracy of the remaining three defects was low. From the diagnosis results of various defects, it can be seen that M, O and P defects had the phenomenon of mutual wrong recognition. This shows that under the diagnosis based on UHF signals, the fault samples of the three types of defects are more susceptible to the strong randomness of PD. The deep residual CNN cannot accurately recognize the corresponding fault defect types, resulting in a large number of misjudgments.

The ultrasonic detection method has a low recognition accuracy for floating electrode defects, because the ultrasonic signal will produce severe signal attenuation through SF_6_ during the propagation process, making the sample characteristics indistinct. The ultrasonic testing method showed excellent recognition results for the discharge defects of free metal particles. The defect of free metal particles is mainly due to the periodic changes of the electric field inside the device due to the metal debris generated by the GIS equipment during the installation process or the switching action process. Such metal particles randomly move or jump between the high-voltage conductor and the low-voltage casing. Since the ultrasonic signal generate during the formation of this defect only propagates through the metal shell of the GIS, the problem of signal attenuation caused by propagation through SF_6_ will not occur.

The overall recognition accuracy based on multi-information fusion recognition reached 97.500%, and the recognition accuracy for various types of defects was greater than 94%. Among them, the recognition accuracy of free metal particle defects has been improved most significantly. For the discharge defects of free metal particles, the diagnostic accuracy of ultrasonic testing method is higher. The UHF detection method can more effectively find the metal tip defects in the GIS, which shows that the two diagnosis methods have higher recognition accuracy and strong complementarity. After fusion, the obtained recognition results are more deterministic, and the recognition accuracy is also improved. It can be seen that the multi-information fusion recognition method can effectively improve the certainty and reliability of the judgment result on the basis of ensuring the reliability of the single feature information recognition result.

### 4.3. Result Comparison with Different Methods

To verify the diagnostic performance of the proposed method, several methods commonly used in PD pattern recognition are compared in this paper. The comparison results are shown in Table 2. In order to avoid the differences between different data sources, the PD data used in the comparison methods were all obtained under the same experimental conditions.

It can be seen from Table 2 that the recognition accuracy of the diagnosis method based on multi-information fusion is higher than that of the method based on single fault information. Since a single fault signal cannot fully reflect the insulation fault information contained in the PD, no matter whether the fault features are based on the form of images or based on statistical feature analysis, the obtained diagnosis results cannot be optimal. In the comparison method based on multi-information fusion, the fault feature is in the form of the extracted PD signal statistical feature parameters, and the effect of PD pattern recognition mainly depends on the selected feature parameters. Manual selection of fault feature parameters cannot fully utilize the advantages of automatic feature extraction by deep learning. The proposed method performs feature extraction on PD signals in the form of pictures. At the same time, deep residual CNN is used as the recognition classifier of PD signal feature information. Compared with ordinary CNN, deep residual CNN has a deeper network structure and can perform deeper feature extraction on images. The multi-information fusion method based on ensemble learning also avoids the misclassification or omission of classification due to conflicts between evidence in the traditional D–S evidence theory fusion, and improves the fusion efficiency. Hence, the proposed method exhibits the best diagnostic performance in PD pattern recognition.

## 5. Conclusions

Based on the fact that the traditional GIS PD pattern recognition method based on single fault signal cannot effectively utilize the rich insulation state information contained in PD, this paper proposes an intelligent diagnosis method based on multi-information ensemble learning for GIS insulation defect diagnosis. First, sufficient UHF signals and ultrasonic signals were obtained from PD experiments, and then an automatic feature extraction method based on deep residual CNN was used to extract discriminative features of the two types of PD signals. Finally, the multi-information ensemble learning technology was used to jointly recognize the typical insulation defects of GIS. The decision-level fusion of UHF features and ultrasonic features by ensemble learning was introduced into GIS PD pattern recognition, which realized an effective comprehensive judgment of insulation defects in GIS and was able to reflect the types of insulation defects more comprehensively. Through the idea of multi-information fusion, the difference of the feature information of the two types of fault signals was combined and complemented, and the common and effective information contained in the recognition of each signal was fully excavated, and more prominent diagnosis results were obtained, thereby improving the accuracy and reliability of pattern recognition. The analysis of the experimental results shows that the recognition results obtained after fusion processing by ensemble learning are more prominent than the single signal feature information, and the recognition accuracy of various insulation defects is greater than 94%. To sum up, the proposed method has better fault feature extraction ability, can more comprehensively reflect the types of insulation defects, and has broad application prospects in the field of GIS PD recognition.

## Figures and Tables

**Figure 1 entropy-24-00954-f001:**
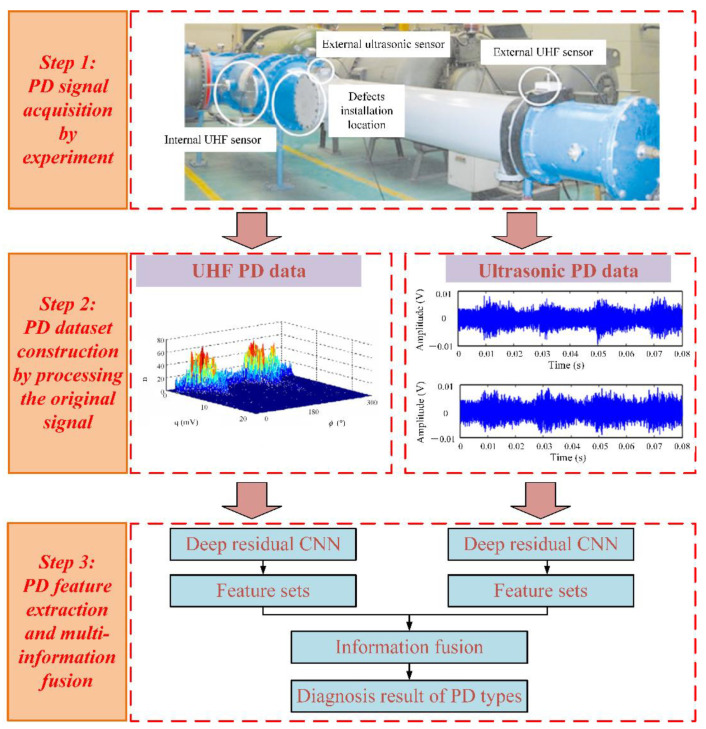
Overall framework of the proposed method.

**Figure 2 entropy-24-00954-f002:**
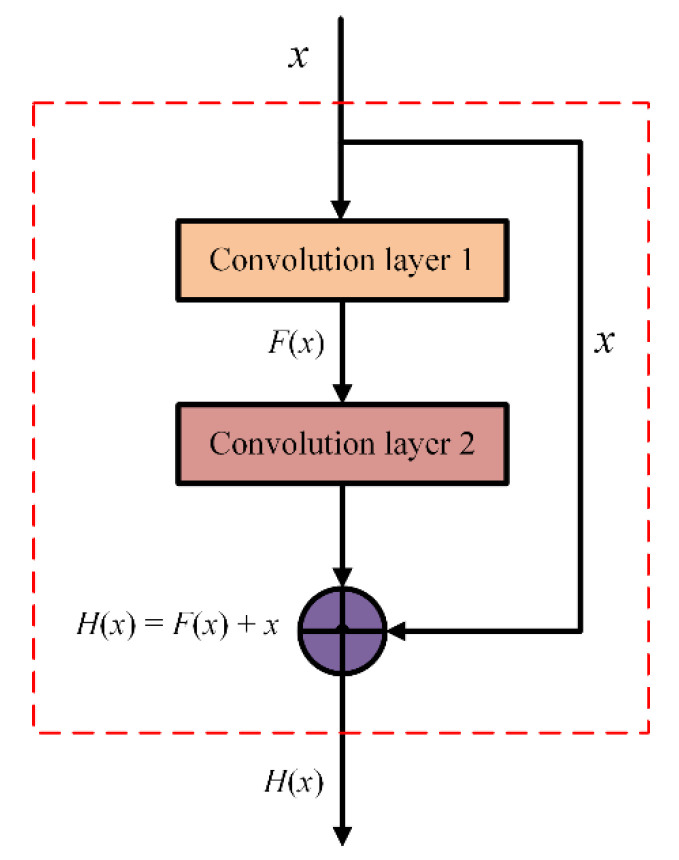
Overall framework of the proposed method.

**Figure 3 entropy-24-00954-f003:**
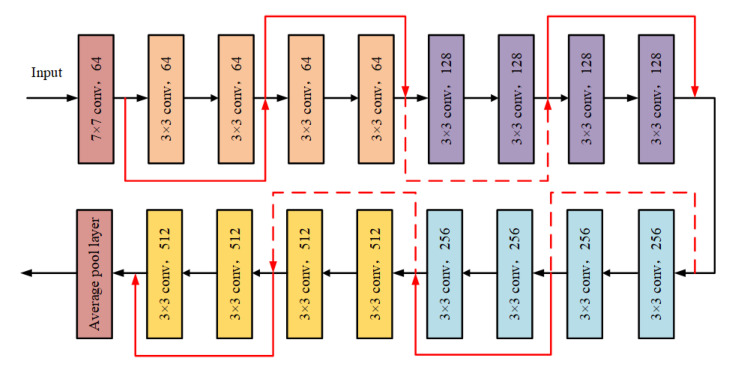
The network structure of the deep residual CNN.

**Figure 4 entropy-24-00954-f004:**
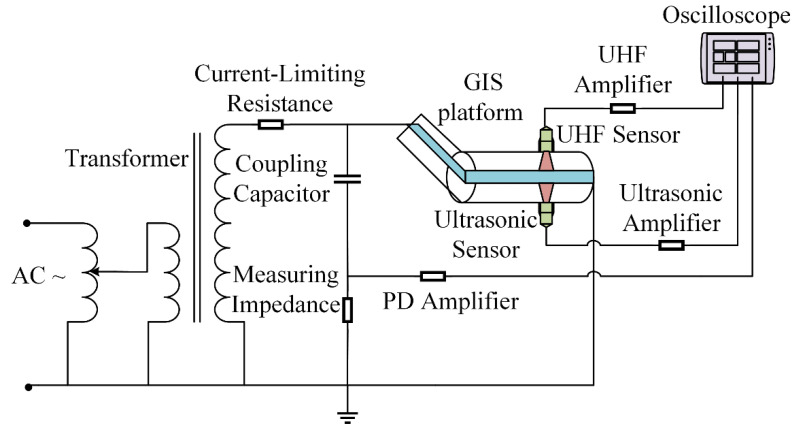
A 252-kV GIS PD experimental schematic diagram.

**Figure 5 entropy-24-00954-f005:**
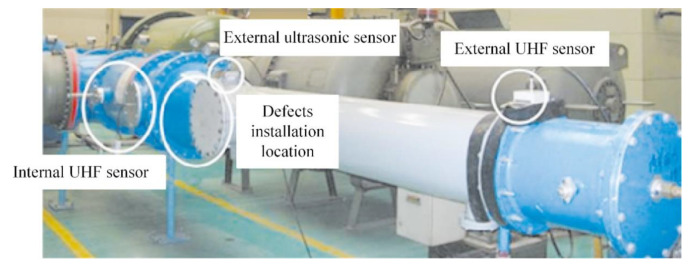
A 252-kV GIS PD test platform.

**Figure 6 entropy-24-00954-f006:**
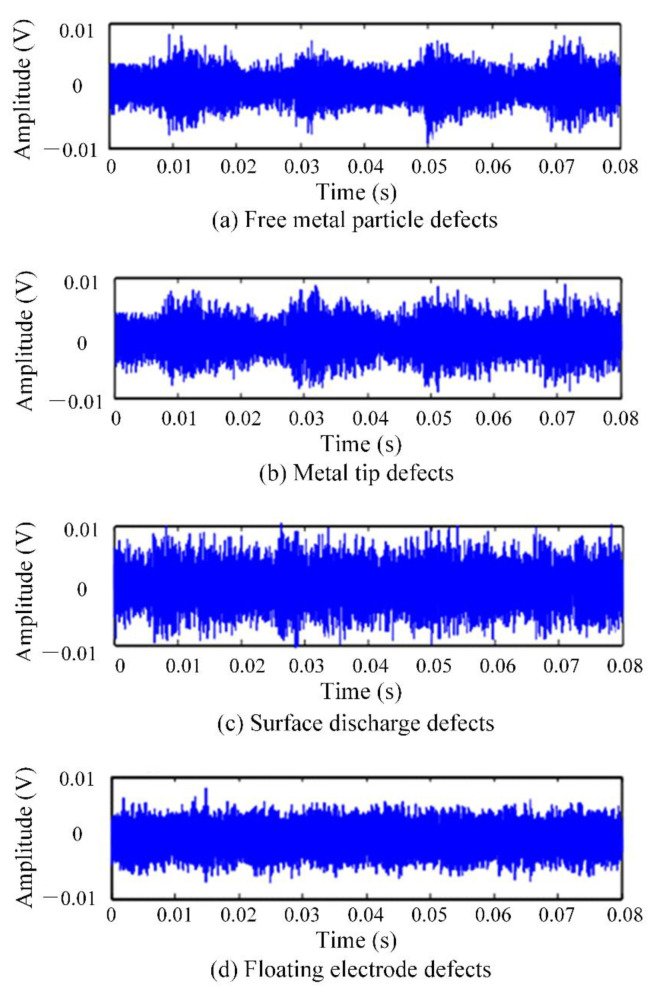
The ultrasonic signals of GIS PD.

**Figure 7 entropy-24-00954-f007:**
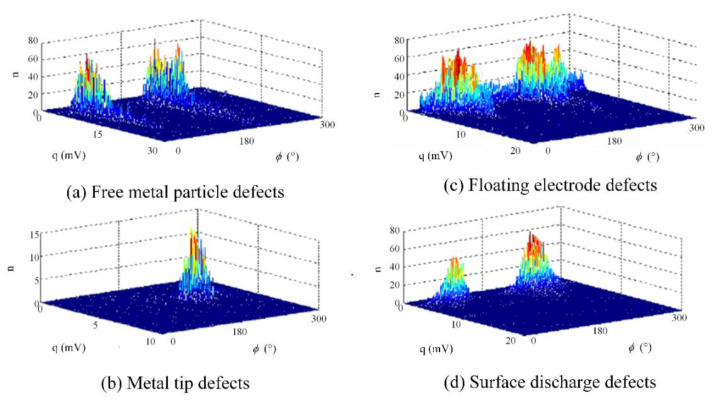
The UHF signals of GIS PD.

**Figure 8 entropy-24-00954-f008:**
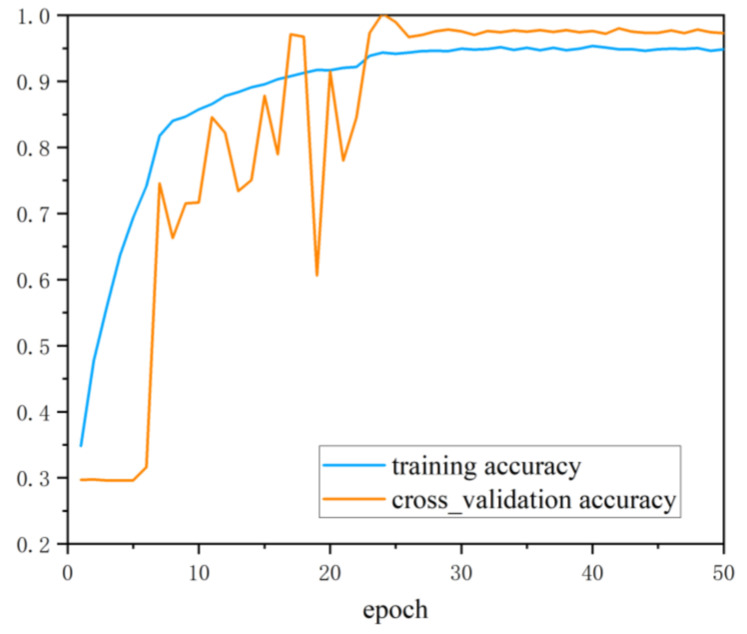
The training accuracy and cross_validation accuracy curves.

**Figure 9 entropy-24-00954-f009:**
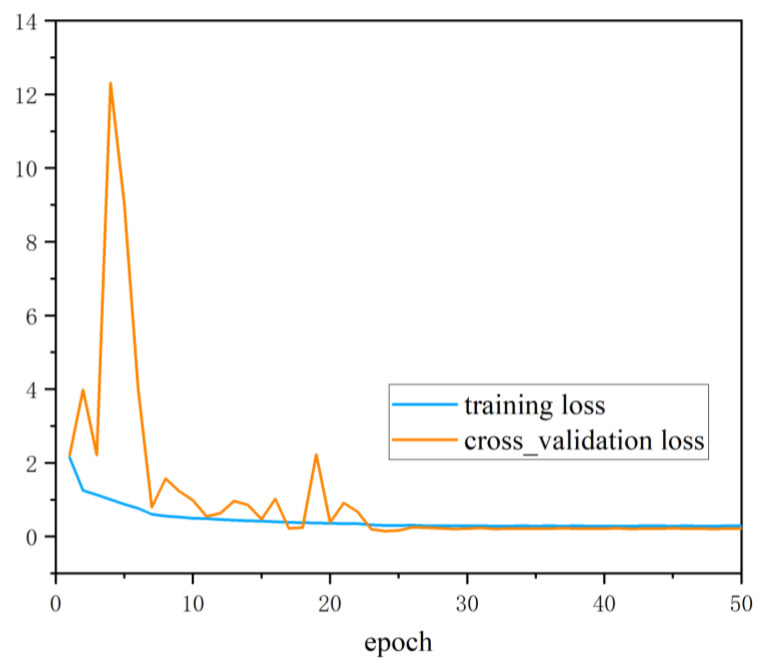
The training loss and cross_validation loss curves.

**Table 1 entropy-24-00954-t001:** The diagnosis result for PD pattern recognition.

Dataset Type	Target Class	Output Class				Overall Accuracy (%)
		M	N	O	P	
UHF	M	173	0	17	10	91.625
	N	0	200	0	0	
	O	12	0	171	17	
	P	3	0	8	189	
Ultrasonic	M	189	7	3	1	88.375
	N	17	167	12	4	
	O	2	2	181	15	
	P	4	7	19	170	
UHF + Ultrasonic	M	198	2	0	0	97.500
	N	0	200	0	0	
	O	4	0	189	7	
	P	3	0	4	193	

**Table 2 entropy-24-00954-t002:** The diagnosis result comparison with different methods.

Ref.	Dataset Type	PD Fault Feature	Classifiers	Accuracy
Tuyet et al. [7]	UHF	phase resolved PD images	Long short-term memory + CNN	93.625%
Ling et al. [23]	UHF	statistical features of phase resolved PD	Support vector machine (SVM)	86.750%
Barrios et al. [24]	UHF	phase resolved PD data	Autoencoder	90.125%
Li L et al. [12]	phase resolved PD + time resolved PD	statistical features of time and frequency domain	BPNN + D-S evidence theory fusion	94.375%
Wu Y et al. [10]	phase resolved PD + time resolved PD + ultrasonic	grayscale image features, statistical features, et al.	SVM + D-S evidence theory fusion	95.125%
Proposed	UHF + ultrasonic	2D images	Deep CNN + D-S evidence theory fusion	95.250%
Proposed	UHF + ultrasonic	statistical features	BPNN + D-S evidence theory fusion	93.750%
Proposed	UHF + ultrasonic	2D images	Deep residual CNN+ ensemble learning	97.500%

## Data Availability

The data presented in this study are available on request from the corresponding author.

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
