# Peer review of "A Novel Method for Pattern Recognition of GIS Partial Discharge via Multi-Information Ensemble Learning"

_entropy, 2022, doi:10.3390/e24070954_

Round 1

Reviewer 1 Report

A novel multi-information ensemble learning for PD pattern recognition is proposed in this work. First, the ultra-high frequency and ultrasonic data of PD under four typical defects of GIS were obtained through experiment. Then the deep residual convolution neural network is used to automatically extract discriminative features. Finally, the multi-information ensemble learning is used to classify PD types at the decision level, which can complement the shortcomings of the independent recognition of the two types of feature information, and has higher accuracy and reliability.

The work is interesting and will be useful to readers in this field. However, several comments need to be addressed a follows:

1. The abstract should also highlight the percentage of the accuracy obtained.

2. Please compare the proposed method with CNN alone using the authors' input and output data.

3. In section 2.2., please show the schematic diagram of the test object used to obtain PD sources.

4. The experiment was conducted under a clean environment. The authors should also evaluate how the work will perform under noise condition, as similar to what is found at the actual site.

5. Include the training loss and accuracy vs. epoch curve, if any.

6. Several related works are missing and shall be cited in the reference. For example:

J.K. Wong, H.A. Illias and A.H.A. Bakar, "A Novel High Noise Tolerance Feature Extraction for Partial Discharge Classification in XLPE Cable Joints," IEEE Transactions on Dielectrics and Electrical Insulation, vol.24, no.1, pp.66-74, February 2017. 

Reviewer 2 Report

Overall, your work is well done and significant. 

Author Response

Thank you for your careful review!

Reviewer 3 Report

1. Multi-information ensemble learning is not addressed clearly. It is suggested to utilize more mathematical formula to form the used machine learning model, especially in ensemble learning.

2.  More comparison of machine learning based models applying to PD problems, especially in the data pre-processing and PD Feature extraction,  should be conducted sufficiently.  PD Feature extraction can refer to the following reference:

C. K. Chang and B. K. Boyanapalli, " Assessment of the insulation status aging in power cable joints using support vector machine" IEEE Transactions on Dielectrics and Electrical Insulation, vol. 28, no. 6, December 2021. 

3. The current form is not suitable for the journal publication. It is suggested to reshape the article and resubmit it again. 

Reviewer 4 Report

Interesting reseach about characterization od PD using multi-information ensemble learning.

Author Response

Thank you for your careful review!

Round 2

Reviewer 3 Report

The authors have done appropriate revision.